

# Chaotic correlation functions with complex fermions

**Ritabrata Bhattacharya[1*], Dileep P. Jatkar[1†] and Arnab Kundu[2‡]**

**1** Harish-Chandra Research Institute, Homi Bhabha National Institute (HBNI)
Chhatnag Road, Jhunsi, Allahabad 211 019, India
**2** Theory Division, Saha Institute of Nuclear Physics, Homi Bhabha National Institute (HBNI)
1/AF Bidhannagar, Kolkata 700064, India

⋆ ritabratabhattacharya@hri.res.in, † dileep@hri.res.in,
‡ arnab.kundu@saha.ac.in

## Abstract

We study correlation functions in the complex fermion SYK model. We focus, specifically, on the $h = 2$ mode which explicitly breaks conformal invariance and exhibits the chaotic behaviour. We numerically explore a fermion six-point OTOC, with two and three real-time folds, respectively. While our approach is expected to yield an early-time chaotic growth, we nevertheless observe a near-maximal value. Following the program of Gross-Rosenhaus, we estimate the triple short time limit of the six point function. Unlike the conformal modes with high values of $h$, the $h = 2$ mode has contact interaction dominating over the planar in the large $q$ limit.

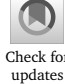
# 1   Introduction

A generic dynamical system is inherently chaotic [1]. For classical systems, chaos can be easily characterized by the sensitivity of trajectories with respect to initial conditions. For quantum systems, lacking in the concept of trajectories, the notion of chaos is more subtle. Often, quantum chaos can be characterized in terms of properties of the spectrum of the Hamiltonian. In the semi-classical approach, there is a relatively simple definition of chaotic behaviour, directly adopted from the sensitivity of classical trajectories with respect to initial conditions.

For classical dynamical systems, characterized by phase space coordinates $\{q(t), p(t)\}$, where $q(t)$ and $p(t)$ are generalized positions and generalized momenta. A particular trajectory is represented by $q(t)$. High sensitivity of the late time trajectory with respect to the initial condition can be quantified as:

$$\exp(\lambda_{\mathrm{L}} t) = \frac{\partial q(t)}{\partial q(0)} \equiv \{q(t), p(t)\} \,, \tag{1}$$

where $\lambda_{\mathrm{L}}$ is the so-called Lyapunov exponent and the right-most expression above is the Poisson bracket [1]. By virtue of the correspondence principle, we obtain a quantum mechanical characterization, by replacing the Poisson bracket with a commutator: $\{q(t), p(t)\} \rightarrow -i\hbar[q(t), p(t)]$ [2]. Instead of computing the commutator, one calculates the squared commutator, so that there is no spurious cancellation due to destructive phases. This argument, however, is limited and does not necessarily imply that allowing for such phases will always cancel the chaotic growth. In this article, we will calculate the cubic power of the commutator, which will explicitly display the exponential growth behaviour.

We define a generic function for the diagnostic of chaos:

$$C_{(n)}(t_1, t_2) \equiv \langle [V(t_1), W(t_2)]^n \rangle \,, \tag{2}$$

where $n \in \mathbb{Z}_+$, and $V$ and $W$ are two self-adjoint operators and the expectation value is defined with respect to a particular state of the system. Note that, in defining the chaos diagnostic in (2), we have recast the chaotic property as a feature of $2n$-point correlation function of the system. In this article, we will explicitly discuss the case for $n = 3$ in a thermal state.

Before doing so, let us briefly look at the $n = 2$ case. Written explicitly, the commutator contains various four-point functions with no particular time-ordering, since $t_1$ and $t_2$ are defined without any ordering. For a thermal state expectation value, using the KMS conditions[1], it is further possible to rearrange the various four-point functions in terms of two pieces: one

---

[1]KMS condition is simply the Euclidean periodicity condition on thermal correlators. For example, for two operators $V(0)$ and $W(t)$, the KMS condition on the two-point function reads:

$$\mathrm{tr}\left(e^{-\beta H} W(t) V(0)\right) = \mathrm{tr}\left(e^{-\beta H} V(0) W(t + i\beta)\right) \,. \tag{3}$$

Here $\beta$ is the inverse temperature. Evidently, this condition can be used to interchange the order of the operators inside a thermal correlator.

time-ordered four-point function and another out-of-time-ordered correlator (OTOC). These are given by $\langle V(0)V(0)W(t)W(t)\rangle$ and $\langle V(0)W(t)V(0)W(t)\rangle$, respectively, choosing $t_1 = 0$ and $t_2 = t$. The time-ordered correlator does not display the exponential growth, it is contained in the four-point OTOC.

For $n = 3$, upon using the KMS condition, the chaos diagnostic in (2) has one time-ordered and two OTOC pieces. These are simply, $\langle V(0)V(0)V(0)W(t)W(t)W(t)\rangle$ (time-ordered) and $\langle V(0)W(t)V(0)W(t)V(0)W(t)\rangle$, $\langle V(0)W(t)V(0)V(0)W(t)W(t)\rangle$, $\langle W(t)V(0)V(0)W(t)V(0)$ $W(t)\rangle$, *etc*, which are OTOC. While a complete understanding of the behaviour of (2) for arbitrary $n$ is desirable, we will explore an exact calculation for $n = 3$ in this article, with a particularly simple model.[2] Note that, the correlator of the form $\langle VWVWVW\rangle$ requires three time-folds in the Schwinger-Keldysh framework, while an OTOC of the form $\langle VWVVWW\rangle$ require only two time-folds. These OTOCs, generically, would not carry the same physical information. For the present model, however, we expect the same four-point OTOC physics in the second case.[3]

The model we consider is a simple generalization of the so-called Sachdev-Ye-Kitaev (SYK) system [3–6], in which one considers fermionic degrees of freedom with an all-to-all interaction. The interaction coupling is drawn from a random Gaussian distribution with a zero mean value and a given width. In the large $N$ limit, in which the number of fermionic degrees of freedom becomes infinite, the system becomes analytically tractable in the sense that the corresponding Schwinger-Dyson equations can be explicitly determined. The solution of this equation readily determines the two-point function, as a function of the coupling strength, in general. In particular, in the low energy limit, this Schwinger-Dyson equation is analytically solvable and yields a two-point function with a manifest $SL(2, R)$ symmetry. In the infra-red (IR), this is described by a conformal field theory (CFT), and the two-point function breaks the conformal group into the $SL(2, R)$ subgroup. In the large $N$ limit, further, the four-point correlator can be explicitly calculated, which yields the corresponding Lyapunov exponent: $\lambda_{\rm L} = 2\pi T$, where $T$ is the temperature of the thermal state. Here, we are working in natural units. This Lyapunov exponent saturates the so-called chaos bound [7]. Intriguingly, the chaos bound saturation also occurs for black holes, in which the local boost factor at the event horizon determines the corresponding Lyapunov exponent as well as the corresponding Hawking temperature. Only extremal black holes have an $SL(2, R)$ global symmetry, due to the existence of an $AdS_2$ sector near the horizon. Correspondingly, the low energy conformal system coming from the SYK model can be shown to capture the essential physics of the $AdS_2$ [8].

The low energy effective action for the SYK model is simply given by a Schwarzian effective action, which can also be shown to arise from the two-dimensional Jackiw-Teitelboim theory in [8,9](In fact, exact derivation of the Schwarzian effective action has been done using the fluid/gravity correspondence [6]). However, in this context, the non-trivial statements of holography necessitates keeping a leading order *correction* away from the purely $AdS_2$ throat, as well as from the purely $CFT_1$ in the IR, hence it goes by the acronym of NAdS/NCFT. From the geometric perspective, $AdS_2$ appears in the following two cases: (i) in the extremal limit of a black hole in asymptotically flat background, (ii) in the deep IR of an asymptotically $AdS_{d+1}$-background. Often, in the second case, the deep IR results from an RG-flow connecting a UV $CFT_d$ to an IR $CFT_1$, as a result of a relevant density perturbation in the UV CFT. Holographically, such operators correspond to turning on a bulk $U(1)$-flux, in the simplest case. A standard example of this is the AdS–Reissner-Nordstrom black hole: It asymptotes to an AdS geometry and the extremal limit consists of an $AdS_2$ in the IR, which is supported by the flux. In the

---

[2]Note that, our explicit expressions of the OTOCs are written in an abuse of notation: KMS conditions will render some of the time-arguments to pick up imaginary parts, proportional to the inverse temperature $\beta$. We suppress these explicit factors here, for simplicity, but take those into account for the explicit calculation of the six-point correlator later.

[3]We thank the Referee for raising this issue.

AdS$_2$ throat, the flux is a simple scalar field, and the boundary theory has a natural notion of a conserved charge and therefore a non-vanishing chemical potential.

The SYK model which is defined with $N$ Majorana fermions, $\psi^i$ ($i = 1, ..., N$) in $(0+1)$ dimensions, cannot have a charge density and correspondingly a non-vanishing chemical potential. If we, instead, consider $N$ complex fermions, $\psi_i$ and $\psi_i^\dagger$, it is natural to introduce a $\psi_i^\dagger \psi_i$ term in the Lagrangian, with chemical potential as the coupling. Similar to the SYK model, we consider a $q$-body interaction term, with $q/2$ number of $\psi$ and $q/2$ number of $\psi^\dagger$, whose couplings are chosen from a Gaussian distribution, with a standard deviation denoted by $J$. In the limit $q \gg 1$, this system becomes exactly solvable in an $(1/q)$-expansion [4], which is what we will use in evaluating the explicit correlation functions. Motivated from the previous paragraph, it is therefore natural to consider a particular generalization of the SYK model with a U(1) global symmetry.

The standard SYK theory, with Majorana fermions, has a particular operator at the conformal fixed point, whose four point OTOC displays chaotic behavior with Lyapunov exponent $\lambda_L = \frac{2\pi}{\beta}$ [3–6, 10–37]. The generalization of this model to complex fermions was done in [38–40]. In [38], the low energy effective action of an SYK-model with complex fermions was discussed. It was shown that the presence of a non-vanishing chemical potential does not break the conformal symmetry in the deep IR, as long as one amends the conformal transformations with a gauge transformation. Thus, the resulting low energy effective action is simply a Schwarzian action along with a free bosonic theory with a standard kinetic term [14, 17]. Therefore, from a strict IR perspective, the maximal chaos holds for any non-vanishing value of the chemical potential.

The non-triviality comes from the order of limits. The exponential growth of the four-point OTOC holds for a larger regime compared to the long-time (and therefore, deep IR) limit. For sufficiently large time, one recovers the maximal chaos. However, there exists an intermediate regime in which the exponential growth takes place with a different Lyapunov exponent. This is physically equivalent to staying in a *medium* energy scale and finding a chaotic behaviour of the correlation function at this energy scale. This associates naturally an RG-flow of the Lyapunov exponent itself.

In the standard SYK model, in the large $q$ limit, the relevant scale in the system is provided by an effective coupling:

$$\mathcal{J}^2 = \frac{q J^2}{2^{q-1}} \, ,$$

which has mass dimension one. The IR CFT resides in the $\mathcal{J} \to \infty$ limit, but the exponential growth of OTOC and subsequently the Lyapunov exponent can be obtained as a perturbation series in $1/\mathcal{J}$. This naturally gives an RG-flow of the Lyapunov exponent [4]. In [40] we studied the SYK model with complex fermions in the large $q$ limit in the presence of a chemical potential $\mu$. Here, in the UV Hamiltonian, we have two natural parameters: $\beta\mu$ and $\beta\mathcal{J}$ and the effective coupling in the IR is given by,

$$\mathcal{J}_{\text{eff}}^2 = \frac{q}{2} \frac{J^2}{(2 + 2\cosh(\mu\beta))^{\frac{q}{2}-1}} \, . \tag{4}$$

The strict IR is located at $\mathcal{J}_{\text{eff}} \to \infty$ limit, and one can calculate systematically the RG-flow of the Lyapunov exponent in a perturbation series in $1/\mathcal{J}_{\text{eff}}$. This RG-flow shows sensitive behaviour for the Lyapunov exponent as the UV parameter $\beta\mu$ is dialled up [40].

In keeping with the theme, in this article, we further compute higher point OTOC for complex fermion SYK-model, with a non-vanishing chemical potential. Our analyses follow closely the analyses in [41], in the large $q$ limit. However, our analyses are performed in the complementary regime in that we completely focus on the operators that display chaotic nature and away from the conformal limit. In spirit of the NAdS/NCFT picture, this is rather

natural regime to consider; in the context of chaotic properties of many body systems, this is an example of a tractable and explicit higher point OTOC which displays the expected exponential growth.

It is worth noting that, even though our calculations are performed with a non-vanishing chemical potential, all analyses are still valid near the conformal point. Therefore, it is not surprising that we find a maximal Lyapunov exponent. On the other hand, our numerical methods are expected to extract a Lyapunov behaviour only at *early times*. We demonstrate, with explicit examples, that such an *early time* calculation still captures a chaotic growth, with a (Schwinger-Keldysh) two-folded 6-point correlator. Furthermore, it captures the maximal Lyapunov with a (Schwinger-Keldysh) three-folded 6-point correlator. At present, we do not have a deeper understanding of an underlying, universal physics that may be responsible for this. However, we do certainly take note that, if generic, this provides us with a tremendous computational convenience in extracting similar chaotic growth, both qualitatively and quantitatively, in various other systems as well.

In this paper, after computing the fermion six point function with a non-vanishing chemical potential, we take the triple short time limit to estimate the the bulk three point correlator, away from the conformal limit. In this regard, we compute bulk three point function(triple short time limit of the fermion six point correlators, neglecting the Schwarzian mode) of the modes satisfying conformal invariance as well as the Schwarzian mode, using the techniques employed by Gross and Rosenhaus [41].

This paper is organized as follows. In Section 2, we briefly review the SYK model with complex fermions. In Section 3, we compute the six point fermion correlator in the triple short time limit. We then interpret it in terms of the bulk three point correlator in the IR limit of the conformal modes and check that we do indeed find them to be of the form of conformal three-point function, in the triple short time limit. We apply this technique in Section 4 to compute the six point function away from the conformal limit and we look at the enhanced contribution due to the non-conformal mode. In Section 5 we take the triple short time limit to determine the three point correlation function of fermion bilinears away from the conformal limit. We carry out this computation in the presence of a chemical potential $\mu$. We also compute the relevant Eucledian correlators which on analytic continuation to real time, is supposed to give us the two fold OTOC as well as the three fold OTOC. From this data we extract the Lyapunov exponent. We conclude with the discussion of our results, and possible future directions.

## 2 SYK model with complex fermions

The SYK model with complex fermion in $0+1$ dimensions is defined by the Hamiltonian with all to all random interaction between $q$ fermions,

$$H = \sum J_{i_1 i_2 \cdots i_{q/2} i_{q/2+1} \cdots i_q} \psi^\dagger_{i_1} \psi^\dagger_{i_2} \cdots \psi^\dagger_{i_{q/2}} \psi_{i_{q/2+1}} \cdots \psi_{i_q} . \tag{5}$$

An exhaustive study of this model is done in [39], we will mention some of the essential features that will be necessary for our analysis. In addition to the higher dimensional operators of the form $\mathcal{O}_n = \frac{1}{N} \sum_i \psi^\dagger_i \partial_t^{2n+1} \psi_i$ which behave in a manner similar to those found in the SYK model with Majorana fermions; we also have the operators of the form $\tilde{\mathcal{O}}_n = \frac{1}{N} \sum_i \psi^\dagger_i \partial_t^{2n} \psi_i$. The lowest lying mode of these operators give the Schwarzian mode and the $U(1)$ charge respectively. In absence of a mass like term in the action the two point function of the particle and anti-particle are the same in the free case as well as the low energy limit of the interacting theory.

$$G_{free}(\tau) = \frac{1}{2} \text{sgn}(\tau), \quad G_c(\tau) = b \frac{\text{sgn}(\tau)}{|\tau|^{\frac{2}{q}}} , \tag{6}$$

where, $G_c(\tau)$ is the propagator in the conformal limit. In the low energy, *i.e.*, IR limit it is possible to obtain the four point function of the fermions using the expansion in the eigenbasis of the quadratic Casimir operator. We will skip the details and only state the results here. Since we have complex fermions, *i.e.*, $\psi_i = \xi_i + \mathbf{i}\eta_i$, in case of the correlation functions we have contributions of two different kind,

$$\langle \psi^\dagger(t_1)\psi(t_2)...\rangle = \langle(\xi(t_1)\xi(t_2)+\eta(t_1)\eta(t_2))...\rangle + \mathbf{i}\langle(\xi(t_1)\eta(t_2)-\eta(t_1)\xi(t_2))...\rangle \,. \tag{7}$$

While the first piece, namely, the real part is anti-symmetric under the exchange of $t_1$ and $t_2$, the second piece is symmetric.

In case of the four point function if we consider the time reversal invariant contribution this leads to two different contributions namely $F^A(\tau_1,\tau_2,\tau_3,\tau_4)$ and $F^S(\tau_1,\tau_2,\tau_3,\tau_4)$ which are respectively anti-symmetric and symmetric under $t_1 \leftrightarrow t_2$ and $t_3 \leftrightarrow t_4$. The first term, *i.e.*, $F^A(\tau_1,\tau_2,\tau_3,\tau_4)$ is identical to the SYK with Majorana fermions but the second term is new and occurs in the complex fermion model. From [39] we have,

$$
\begin{aligned}
\frac{F^A(\tau_1,\tau_2,\tau_3,\tau_4)}{G(\tau_{12})G(\tau_{34})} &= \alpha_0 \int_0^\infty \frac{s\,ds}{\pi^2} \frac{k^A(\frac{1}{2}+is)}{\coth(\pi s)(1-k^A(\frac{1}{2}+is))}\Psi^A_{\frac{1}{2}+is}(\chi) \\
&\quad +\alpha_0 \sum_{2j>0} \frac{2j-\frac{1}{2}}{\pi^2}\frac{k^A(2j)}{1-k^A(2j)}\Psi^A_{2j}(\chi)\,,
\end{aligned}
\tag{8}
$$

$$
\begin{aligned}
\frac{F^S(\tau_1,\tau_2,\tau_3,\tau_4)}{G(\tau_{12})G(\tau_{34})} &= \alpha_0 \int_0^\infty \frac{s\,ds}{\pi^2} \frac{k^S(\frac{1}{2}+is)}{\coth(\pi s)(1-k^S(\frac{1}{2}+is))}\Psi^S_{\frac{1}{2}+is}(\chi) \\
&\quad +\alpha_0 \sum_{2j+1>0} \frac{2j+\frac{1}{2}}{\pi^2}\frac{k^S(2j+1)}{1-k^S(2j+1)}\Psi^S_{2j+1}(\chi)\,,
\end{aligned}
\tag{9}
$$

where,

$$\chi = \frac{\tau_{12}\tau_{34}}{\tau_{13}\tau_{24}}\,, \tag{10}$$

is the conformal cross ratio and $\Psi^A$ and $\Psi^S$ are linear combinations of the eigen-functions of the quadratic Casimir. $\Psi^A$ is antisymmetric, while $\Psi^S$ is symmetric under the transformation,

$$\chi \to \frac{\chi}{\chi-1}\,, \tag{11}$$

which effectively exchanges the first two or last two arguments of four point function. Finally $k^A$ and $k^S$ are eigenvalues of the retarded kernels (for antisymmetric and symmetric) which commute with the Casimir.

## 2.1 Large $q$ limit

We now augment this $q$-point interaction with a quadratic coupling term by introducing a chemical potential $\mu$ which couples to the conserved charge $\sum_i \psi^\dagger_i \psi_i$. The fermion propagator in the Fourier space derived from the quadratic part of the action is,

$$G_0(\mu,\omega) = \frac{1}{i\omega+\mu}\,. \tag{12}$$

Once we take the interaction terms in the Hamiltonian into account it gives the dressed propagator. In the large $N$ limit, the contribution of melonic diagrams dominates. If we also take large $q$ limit ($q < N$) then the loop corrected propagator is amenable to analytic computations,

$$G(\mu,\tau) = G_0(\mu,\tau)\left(1+\frac{g(\mu,\tau)}{q}+\cdots\right)\,, \tag{13}$$

where, $G_0(\mu, \tau)$ is the free propagator in real space. To compute the two-point function in the interacting theory one sets up the Schwinger-Dyson equation and seeks a solution in the large $q$ limit. This Schwinger-Dyson equation at finite inverse temperature $\beta$ can be cast in the form of a differential equation,

$$(\partial_\tau - \mu)^2 [G_0(\mu, \tau)g(\mu, \tau)] = \frac{qJ^2 G_0(\mu, \tau)}{(2 + 2\cosh(\mu\beta))^{q/2-1}} \exp\left(\frac{1}{2}\{g(\mu, \tau) + g(\mu, -\tau)\}\right). \quad (14)$$

The solution to this equation in given by [40],

$$e^{g(\mu, \tau)} = \frac{\cos^2\left(\frac{\pi\nu}{2}\right)}{\cos^2\left(\pi\nu\left(\frac{|\tau|}{\beta} - \frac{1}{2}\right)\right)}, \quad \text{where} \quad \beta\mathcal{J}_{\text{eff}} = \frac{\pi\nu}{\cos\left(\frac{\pi\nu}{2}\right)}, \quad (15)$$

where $\mathcal{J}_{\text{eff}}$ is defined in (4).

## 3 Correlation Functions

Let us begin with the short time, *i.e.*, $\tau_1 - \tau_2 = \tau_{12} \to 0$ limit of the four point function both for the symmetric and anti-symmetric case,

$$F^A(\tau_1, \tau_2, \tau_3, \tau_4) = G(\tau_{12})G(\tau_{34}) \sum_{n=1}^{\infty} c_n^2 \left(\frac{|\tau_{12}\tau_{34}|}{|\tau_{13}\tau_{14}|}\right)^{h_n},$$

$$F^S(\tau_1, \tau_2, \tau_3, \tau_4) = G(\tau_{12})G(\tau_{34}) \sum_{n=1}^{\infty} \tilde{c}_n^2 \left(\frac{|\tau_{12}\tau_{34}|}{|\tau_{13}\tau_{14}|}\right)^{h_n}. \quad (16)$$

When we calculate the the six point function of the complex fermions we go to different short time limits, where the correlation functions take some effective form. In the triple short time limit we calculate it as an effective three point function of the fermion bi-linear operators. This way one can compute the correlation function near points where different arguments approach each other yielding poles and by the property of being analytic everywhere else we get the full contribution.

In the remaining part of this article we calculate the $O(1/N^2)$ coefficient of the six point function with respect to the $1/N$ expansion. To this order there are contributions from the contact diagrams as well as from the planar diagrams (see Fig.1).

We will now write down the corresponding expressions:

$$\mathcal{S} = \mathcal{S}_1 + \mathcal{S}_2 + \tilde{\mathcal{S}}_1 + \tilde{\mathcal{S}}_2, \quad (17)$$

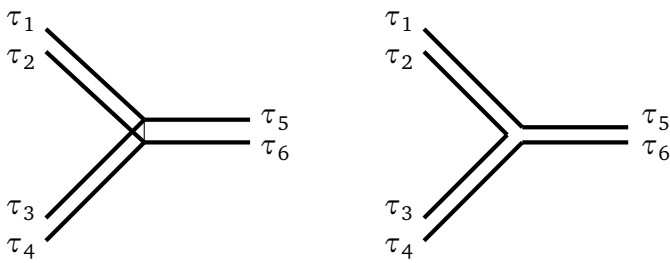

Figure 1: Contact (left) and Planar (right) diagrams for six point function. The double lines are dressed four point functions which include melonic corrections.

where the contributions of $\mathcal{S}_1$ (contact) and $\mathcal{S}_2$ (planar) are exactly same as those in [41], *i.e.*, they agree with corresponding results for the Majorana fermions. In case of the SYK model with complex fermions, if we demand time reversal invariance, (since the Hamiltonian is itself time reversal invariant) we have only two other contributions. Now,

$$\frac{\tilde{\mathcal{S}}_1}{15} = (q-1)(q-2)J^2 \iint_{-\infty}^{\infty} d\tau_a d\tau_b G(\tau_{ab})^{q-3} F^S(\tau_1, \tau_2, \tau_a, \tau_b) \times$$
$$F^A(\tau_3, \tau_4, \tau_a, \tau_b)F^S(\tau_5, \tau_6, \tau_a, \tau_b)\,, \tag{18}$$

is the contact diagram contribution. Here, we have written only one particular assignment of the arguments; there are other possible assignments whose contributions account for the factor of $1/15$ on the left hand side. There are total 15 possible independent configurations[4].

We will use same symbol $h$ to denote the conformal weight of the bi-linear operators both for $F^A$ and $F^S$, although the values are different for the two: for $F^A \Rightarrow h_n = 2n+1+2\Delta+O\left(\frac{1}{k}\right)$, and for $F^S \Rightarrow h_n = 2n + 2\Delta + O\left(\frac{1}{k}\right)$. The planar contribution is given by,

$$\frac{\tilde{\mathcal{S}}_2}{15} = \int_{-\infty}^{\infty} d\tau_a d\tau_b d\tau_c \, \mathcal{F}_{amp}^S(\tau_1, \tau_2, \tau_a, \tau_b)\mathcal{F}_{amp}^A(\tau_3, \tau_4, \tau_b, \tau_c)\mathcal{F}_{amp}^S(\tau_5, \tau_6, \tau_c, \tau_a)\,, \tag{19}$$

where,

$$\mathcal{F}_{amp}^S(\tau_1, \tau_2, \tau_3, \tau_4) = J^2 \int d\tau_0 F^S(\tau_1, \tau_2, \tau_3, \tau_0)G(\tau_{40})^{q-1}\,. \tag{20}$$

Using the Selberg integrals in its special and general forms, one obtains:

$$\mathcal{F}_{amp}^S(\tau_1, \tau_2, \tau_3, \tau_4) = G(\tau_{12}) \sum_n \tilde{c}_n^2 \tilde{\xi}_n sgn(\tau_{12})sgn(\tau_{43}) \frac{|\tau_{12}|^{h_n}|\tau_{34}|^{h_n-1}}{|\tau_{24}|^{h_n+1-2\Delta}|\tau_{23}|^{h_n-1+2\Delta}}\,. \tag{21}$$

Using the short time expansion of four point amplitudes, we get:

$$\frac{\tilde{S}_1}{15} = b^q(q-1)(q-2)J^2 \sum_{n,m,k} \tilde{c}_n c_m \tilde{c}_k |\tau_{12}|^{h_n}|\tau_{34}|^{h_m}|\tau_{56}|^{h_k} G(\tau_{12})G(\tau_{34})G(\tau_{56})I_{nmk}^{(1)}\,,$$

$$\frac{\tilde{S}_2}{15} = b^q(q-1)(q-2)J^2 \sum_{n,m,k} \tilde{c}_n c_m \tilde{c}_k \tilde{\xi}_n \xi_m \tilde{\xi}_k |\tau_{12}|^{h_n}|\tau_{34}|^{h_m}|\tau_{56}|^{h_k} \times \tag{22}$$
$$G(\tau_{12})G(\tau_{34})G(\tau_{56})I_{nmk}^{(2)}\,,$$

where explicit expressions of the constants $c$, $\xi$, $\tilde{c}$, $\tilde{\xi}$ are given in appendix A. The integrals $I^{(1)}$ and $I^{(2)}$ are given by,

$$I_{nmk}^{(1)}(\tau_1, \tau_3, \tau_5) = sgn(\tau_{12})sgn(\tau_{56}) \int_{-\infty}^{\infty} d\tau_a d\tau_b \frac{sgn(\tau_{1a}\tau_{1b}\tau_{5a}\tau_{5b})|\tau_{ab}|^{h_n+h_m+h_k-2}}{|\tau_{1a}|^{h_n}|\tau_{1b}|^{h_n}|\tau_{3a}|^{h_m}|\tau_{3b}|^{h_m}|\tau_{5a}|^{h_k}|\tau_{5b}|^{h_k}}\,, \tag{23}$$

$$I_{nmk}^{(2)}(\tau_1, \tau_3, \tau_5) = -sgn(\tau_{12})sgn(\tau_{56}) \int_{-\infty}^{\infty} d\tau_a d\tau_b d\tau_c \left[ \frac{sgn(\tau_{3b})sgn(\tau_{3c})}{|\tau_{3c}|^{h_m-1+2\Delta}|\tau_{3a}|^{h_m+1-2\Delta}} \times \right.$$
$$\left. \frac{sgn(\tau_{ab})sgn(\tau_{bc})|\tau_{ab}|^{h_n-1}|\tau_{ca}|^{h_m-1}|\tau_{bc}|^{h_k-1}}{|\tau_{1a}|^{h_n-1+2\Delta}|\tau_{1b}|^{h_n+1-2\Delta}|\tau_{5b}|^{h_k-1+2\Delta}|\tau_{5c}|^{h_k+1-2\Delta}} \right]\,. \tag{24}$$

---

[4]Instead of taking $\tau_1, \tau_2$ for the first four point function argument we could take any other two, for example $\tau_1, \tau_3$. Also interchanging the vertices among themselves This way the total number of possibilities is given by $\binom{6}{2} \times \binom{4}{2} \times \frac{1}{3!} = 15$

The integral (23) can be simplified by the change of variables, $\tau_a \to \tau_1 - (1/\tau_a)$, and $\tau_b \to \tau_1 - (1/\tau_b)$. The simplification is done by first decomposing the integral into sums of integrals. Namely the integration from $-\infty$ to $\infty$ will be written as a sum of two, an integral from $-\infty$ to 0 and an integral from 0 to $\infty$. We implement the change of variables on each fragment separately, simplify each of them before recombining them back. At the end of this exercise we get,

$$
I_{nmk}^{(1)}(\tau_1, \tau_3, \tau_5) = \text{sgn}(\tau_{12})\text{sgn}(\tau_{56}) \int_{-\infty}^{\infty} d\tau_a \, d\tau_b \left[ \frac{1}{|\tau_{31}|^{2h_m}|\tau_{51}|^{2h_k}} \times \right.
$$
$$
\left. \frac{|\tau_{ab}|^{h_n+h_m+h_k-2}\text{sgn}(\tau_{51}\tau_a + 1)\text{sgn}(\tau_{51}\tau_b + 1)}{|\tau_a + \frac{1}{\tau_{31}}|^{h_m}|\tau_b + \frac{1}{\tau_{31}}|^{h_m}|\tau_a + \frac{1}{\tau_{51}}|^{h_k}|\tau_b + \frac{1}{\tau_{51}}|^{h_k}} \right] .
$$
(25)

These change of variables are followed up by another pair of change of variables which are carried out in a sequential manner. We will first implement $\tau_a \to \tau_a - (1/\tau_{31})$, and $\tau_b \to \tau_b - (1/\tau_{31})$ and then we will rescale the integration variables $\tau_a \to (\tau_{53}\tau_a)/(\tau_{31}\tau_{51})$ and $\tau_b \to (\tau_{53}\tau_b)/(\tau_{31}\tau_{51})$.

$$
I_{nmk}^{(1)}(\tau_1, \tau_3, \tau_5) = \frac{\text{sgn}(\tau_{12})\text{sgn}(\tau_{56})}{|\tau_{31}|^{h_n+h_m-h_k}|\tau_{51}|^{h_n+h_k-h_m}|\tau_{53}|^{h_k+h_m-h_n}} \times \tilde{I}_{nmk}^{(1)}(h_n, h_m, h_k) ,
$$

$$
\tilde{I}_{nmk}^{(1)}(h_n, h_m, h_k) = \int_{-\infty}^{\infty} d\tau_a \, d\tau_b \frac{|\tau_{ab}|^{h_n+h_m+h_k-2}\text{sgn}(\tau_a - 1)\text{sgn}(\tau_b - 1)}{|\tau_a|^{h_m}|\tau_b|^{h_m}|1-\tau_a|^{h_k}|1-\tau_b|^{h_k}} = \tilde{S}_{2,2}^{full}(\alpha, \beta, \gamma) ,
$$
(26)

where, $\alpha = -h_n + 1$, $\beta = -h_k + 1$, and $\gamma = \frac{h_n+h_m+h_k}{2} - 1$. To summarize, the aim of the above exercise was to obtain a conformal three point correlation function for fermion bi-linear operators denoted above by $I_{nmk}^{(1)}$.

As in [41], we divide the Selberg integral, $\tilde{S}_{2,2}^{full}$, into different parts. This is achieved by decomposing the integral into three pieces $[-\infty, 0]$, $[0, 1]$ and $[1, \infty]$ for each integration variable. This results in six Selberg integrals with appropriately modified arguments. Carefully keeping track of the signs gives,

$$
\tilde{S}_{2,2}^{full}(\alpha, \beta, \gamma) = S_{2,2}(\alpha, \beta, \gamma) + S_{2,2}(1 - \alpha - \beta - 2\gamma, \beta, \gamma)
$$
$$
+ S_{2,2}(1 - \alpha - \beta - 2\gamma, \alpha, \gamma) + 2S_{2,1}(1 - \alpha - \beta - 2\gamma, \alpha, \gamma)
$$
$$
- 2S_{2,1}(\alpha, \beta, \gamma) - 2S_{2,1}(\alpha, 1 - \alpha - \beta - 2\gamma, \gamma) .
$$
(27)

The generalized Selberg integrals and some important results which are used above are given in [41], but for completeness we give the relevant definitions here,

$$
S_{n,n}(\alpha, \beta, \gamma) = \int_{[0,1]^n} d\tau_1 ... d\tau_n \prod_{i=1}^{n} |\tau_i|^{\alpha-1}|1 - \tau_i|^{\beta} \prod_{i<j} |\tau_{ij}|^{2\gamma} ,
$$
$$
S_{n,p}(\alpha, \beta, \gamma) = \int_{[0,1]^p} \int_{[1,\infty)^{n-p}} d\tau_1 ... d\tau_n \prod_{i=1}^{n} |\tau_i|^{\alpha-1}|1 - \tau_i|^{\beta} \prod_{i<j} |\tau_{ij}|^{2\gamma} .
$$
(28)

In a similar fashion one can manipulate $I_{nmk}^{(2)}$ to bring it in a form of the conformal three point function. This computation, however, is considerably more involved, we instead do the analysis in the large $q$ limit. The $I_{nmk}^{(2)}$ in our case differs from that obtained in [41] by only

the $sgn$ functions while the rest of the integrand has exactly the same form. So for us also at large $q$, $I^{(2)}_{nmk}$ takes the form,

$$I^{(2)}_{nmk}(\tau_1, \tau_2, \tau_3) \approx \frac{\tilde{s}^{(2)}_{nmk}}{|\tau_{31}|^{h_n+h_m-h_k}|\tau_{51}|^{h_n+h_k-h_m}|\tau_{53}|^{h_k+h_m-h_n}} + \cdots \tag{29}$$

In our case of course $\tilde{s}^{(2)}_{nmk}$ is different from $s^{(2)}_{nmk}$ obtained by Gross and Rosenhaus [41].

## 4 Away from the Conformal Limit

In this section we carry out the calculation of correlation functions away from the conformal IR fixed point. In our earlier work [40], we studied the effect of introducing a chemical potential $\mu$, in the SYK-model with complex fermions. We found that a non-zero $\mu$ takes us away from the conformal limit since it explicitly introduces a scale in the problem. The effect of introduction of this scale parameter is reflected in the chaotic behavior of the model, namely, it brings down the value of the Lyapunov exponent. We computed the required quantities and studied the maximally chaotic mode(in the large $q$ limit where things could be handled analytically).

We write below the relevant expressions in the large $q$ limit. The two point function(to the leading order in large $N$) in the region $\tau > 0$ is given by,

$$G(\mu, \tau) = G_0(\mu, \tau)\left(1 + \frac{1}{q}\log\left(\frac{\cos\left(\frac{\pi\nu}{2}\right)}{\cos\left[\pi\nu\left(\frac{1}{2} - \frac{\tau}{\beta}\right)\right]}\right) + ..\right), \tag{30}$$

where,

$$G_0(\mu, \tau) = -\frac{e^{\mu\tau}}{e^{\mu\beta}+1}, 0 \leq \tau \leq \beta, \tag{31}$$

$$G_0(\mu, \tau) = \frac{e^{\mu\tau}}{e^{-\mu\beta}+1}, -\beta \leq \tau \leq 0. \tag{32}$$

The above relation can be written in a compact manner by writing,

$$G_0(\mu, \tau) = -sgn(\tau)\frac{e^{\mu\tau}}{e^{\mu\beta sgn(\tau)}+1}, \quad 0 \leq \tau \leq \beta. \tag{33}$$

We now aim at calculating the enhanced contribution to the four point function slightly away from the conformal limit with the chemical potential $\mu$. Note that since we want to be slightly away from the IR, we will keep $\mu\beta$ to be small and expand all functions in this variable. Then it can be interpreted that we move slightly away from the IR by turning on a small chemical potential.

To this end we need to first calculate the shift in the eigenvalue of the Kernel for the $h = 2$ mode. For this we incorporate the technique used in [4], we begin with the equation,

$$K\Psi = k\Psi, \quad \Rightarrow \quad \int\int K(\tau_1, \tau_2, \tau_3, \tau_4)\Psi(\tau_3, \tau_4)d\tau_3 d\tau_4 = k\Psi(\tau_1, \tau_2). \tag{34}$$

The Kernel is given by,

$$K(\tau_1, \tau_2, \tau_3, \tau_4) = -(-1)^{q/2}J^2(q-1)G(\mu, \tau_{13})G(\mu, -\tau_{24})G(\mu, \tau_{34})^{q/2-1}G(\mu, -\tau_{34})^{q/2-1}. \tag{35}$$

We will work in the large $q$ limit. Substituting the Kernel in equation (34) gives,

$$-qJ^2 \int d\tau_3 d\tau_4 \frac{sgn(\tau_{13})sgn(\tau_{24})e^{\mu\tau_{13}}e^{-\mu\tau_{24}}}{(e^{\mu\beta sgn(\tau_{13})}+1)(e^{-\mu\beta sgn(\tau_{24})}+1)} \frac{\cos^2\left(\frac{\pi\nu}{2}\right)}{\sin^2\left(\frac{\tilde{x}_{34}}{2}\right)} \times$$

$$\frac{\Psi(\tau_3,\tau_4)}{\{(e^{\mu\beta sgn(\tau_{34})}+1)(e^{-\mu\beta sgn(\tau_{34})}+1)\}^{q/2-1}} = k\Psi(\tau_1,\tau_2), \qquad (36)$$

where $\nu$ is defined in (15) and $\tilde{x}_{ij} = \frac{2\pi\nu\tau_{ij}}{\beta} + \pi(1-\nu)$. Multiplying (36) by $e^{-\mu\tau_{12}}$ on both sides of the equation, and differentiating twice, once with respect to $\tau_1$ and once with respect to $\tau_2$ gives,

$$\partial_{\tau_1}\partial_{\tau_2}\left(\frac{sgn(\tau_{13})sgn(\tau_{24})}{(e^{\mu\beta sgn(\tau_{13})}+1)(e^{-\mu\beta sgn(\tau_{24})}+1)}\right) = 4\delta(\tau_{13})\delta(\tau_{24}). \qquad (37)$$

Using the parametrization $k = \frac{2}{h(h-1)}$ eq.(36) reduces to,

$$-\frac{qJ^2\cos^2\left(\frac{\pi\nu}{2}\right)}{(2+2\cosh(\mu\beta))^{q/2-1}} \times \frac{e^{-\mu\tau_{12}}}{\sin^2\left(\frac{\tilde{x}_{12}}{2}\right)}\Psi(\tau_1,\tau_2) = \frac{2}{h(h-1)}\partial_{\tau_1}\partial_{\tau_2}\left(e^{-\mu\tau_{12}}\Psi(\tau_1,\tau_2)\right). \qquad (38)$$

If we substitute $\Psi(\tau_1,\tau_2) = e^{\mu\tau_{12}}e^{-in(\tau_1+\tau_2)}\psi_n(\tau_{12})$ followed by some manipulation of eq.(38) (also using (15)) we arrive at the differential equation,

$$\left[n^2 + 4\partial_x^2 - \frac{\nu^2 h(h-1)}{\sin^2\left(\frac{\tilde{x}}{2}\right)}\right]\psi_n(x) = 0. \qquad (39)$$

Here, $x = \frac{2\pi\tau}{\beta}$ and we have suppressed the subscript on $\tau$ since everything is now a function of the time difference $\tau_{12}$.

The solution to this equation with appropriate boundary condition is well known. In fact this is the same equation as obtained in [4]. The solution is given by, (with $\tilde{n} = n/\nu$)

$$\psi_{h,n}(x) = \left(\sin\frac{\tilde{x}}{2}\right)^h {}_2F_1\left(\frac{h-\tilde{n}}{2}, \frac{h+\tilde{n}}{2}, \frac{1}{2}; \cos^2\left(\frac{\tilde{x}}{2}\right)\right), \qquad n = \text{even},$$

$$\psi_{h,n}(x) = \cos\frac{\tilde{x}}{2}\left(\sin\frac{\tilde{x}}{2}\right)^h {}_2F_1\left(\frac{h-\tilde{n}+1}{2}, \frac{h+\tilde{n}+1}{2}, \frac{3}{2}; \cos^2\left(\frac{\tilde{x}}{2}\right)\right), \qquad n = \text{odd}.$$

The quantization condition on $h$ is obtained by demanding that the wave function vanishes at $x = 0$, i.e., $\tilde{x} = \pi(1-\nu)$. As we approach the conformal limit $\nu \to 1$ this solution actually diverges for generic values of $h$ near 2 (we are interested in the $h = 2$ eigenfunctions). But we want values of $h$ such that the solutions are finite or vanishing, so the first or second argument of the hypergeometric function has to be a negative integer. This gives the quantization of $h$ near 2 to be,

$$h_n = 2 + |\tilde{n}| - |n|, \quad h_n = 2 + |n|\left(\frac{1-\nu}{\nu}\right). \qquad (40)$$

This gives the shift in the eigenvalue $k = \frac{2}{h(h-1)}$ to be,

$$k(2,n) = 1 - \frac{3|n|}{2}(1-\nu) + \left(\frac{7n^2}{4} - \frac{3|n|}{2}\right)(1-\nu)^2 + \dots \qquad (41)$$

This result is identical to the shift obtained in [4], only difference being that $\nu$ now depends on the effective coupling $\beta\mathcal{J}_{\text{eff}}$ instead of $\beta\mathcal{J}$.

## 4.1 The enhanced four point contribution

Let us now look at the four point function and use the above result to figure out the enhanced contribution for the Schwarzian mode slightly away from the conformal limit. We begin with the expansion of the four point function in the basis of eigenfunctions of the Kernel (using the variable $\theta = \frac{2\pi\tau}{\beta}$ on the thermal circle and the period becomes $2\pi$),

$$\frac{\mathcal{F}(\theta_1, \theta_2, \theta_3, \theta_4)}{G(\theta_{12})G(\theta_{34})} = 2 \sum_{h,n} \frac{k(h,n)}{1-k(h,n)} \Psi_{h,n}^{\text{exact}}(\theta_1, \theta_2)\Psi_{h,n}^{\text{exact}*}(\theta_3, \theta_4). \tag{42}$$

To find the enhanced contribution of the Schwarzian or $h = 2$ mode we use the eigenfunction of the Casimir for $h = 2$ and the shifted eigenvalue in the denominator. In the numerator we just use the eigenvalue with $h = 2$ in the IR. This is done to ensure that we are only slightly away from the conformal limit driven by introducing a small chemical potential. Here we will use all results for the large $q$ limit,

$$\begin{aligned}\frac{\mathcal{F}(\theta_1, \theta_2, \theta_3, \theta_4)}{G(\theta_{12})G(\theta_{34})} &= \frac{4\beta\mathcal{J}}{\pi^2} \sum_{|n|\geq 2} \frac{e^{in(y'-y)}}{n^2(n^2-1)(1+a)}\left[\frac{\sin\left(\frac{nx}{2}\right)}{\tan\left(\frac{x}{2}\right)} - n\cos\left(\frac{nx}{2}\right)\right] \times \\ &\qquad\qquad \left[\frac{\sin\left(\frac{nx'}{2}\right)}{\tan\left(\frac{x'}{2}\right)} - n\cos\left(\frac{nx'}{2}\right)\right],\end{aligned} \tag{43}$$

where,

$$x = \theta_1 - \theta_2, \quad x' = \theta_3 - \theta_4, \quad y = \frac{\theta_1 + \theta_2}{2}, \quad y' = \frac{\theta_3 + \theta_4}{2}, \quad a = \left(\frac{q}{2}-1\right)\frac{(\mu\beta)^2}{8}, \tag{44}$$

and for large $\beta\mathcal{J}_{\text{eff}}$ we have used $1 - \nu \sim \frac{2}{\beta\mathcal{J}_{\text{eff}}} \approx \frac{2}{\beta\mathcal{J}} + \left(\frac{q}{2}-1\right)\frac{(\mu\beta)^2}{4\beta\mathcal{J}}$.

We will now carry out the sum over $n$. The final expression after all simplifications can be written in a compact form [4]. To proceed with the six point function using these results we use numerical computation, in carrying out the integration. Subsequently, we extract the chaotic behavior of the OTO six point correlator, even though we do not have an analytic result. This will be discussed in a later section.

## 4.2 The Contact and the Planar diagrams

There are two types of topologies of Feynman diagrams contributing to six point function. The contact diagrams and the planar diagrams (see Fig. 1). We will now claim that among these diagrams, which contribute to the six point function at the leading order in $\sim 1/N$, the contribution of the contact diagrams dominates over that of the planar ones by an order $q^4$, for the enhanced non-conformal mode contribution to the four point function. So at large $q$, the contact diagrams dominate over the planar ones, and hence we will look at only the former. But before that we will demonstrate this dominance of the contact diagrams below by studying the scaling behaviour in the large $q$ limit of the contact and planar contributions.

The contact contribution is given by,

$$\begin{aligned}S_c = (q-1)(q-2)J^2 \int d\tau_a d\tau_b G(\tau_{ab})^{\frac{q}{2}-3}G(-\tau_{ab})^{\frac{q}{2}} \times \\ \mathcal{F}(\tau_1, \tau_2, \tau_a, \tau_b)\mathcal{F}(\tau_3, \tau_4, \tau_a, \tau_b)\mathcal{F}(\tau_5, \tau_6, \tau_a, \tau_b).\end{aligned} \tag{45}$$

Whereas the planar contribution is,

$$\mathcal{S}_p = \int_{-\infty}^{\infty} d\tau_a d\tau_b d\tau_c \; \mathcal{F}_{\text{amp}}(\tau_1, \tau_2, \tau_a, \tau_b)\mathcal{F}_{\text{amp}}(\tau_4, \tau_3, \tau_c, \tau_a)\mathcal{F}_{\text{amp}}(\tau_5, \tau_6, \tau_b, \tau_c), \tag{46}$$

where,

$$\mathcal{F}_{\text{amp}}(\tau_1, \tau_2, \tau_3, \tau_4) = -\int_0^\beta d\tau_0 \mathcal{F}(\tau_1, \tau_2, \tau_3, \tau_0) \int \frac{d\omega_4}{2\pi} e^{-i\omega_4 \tau_{40}} \frac{1}{G(\mu, \omega_4)},$$

is the amputated four point function. We can use the SD equations to write,

$$\frac{1}{G(\mu, \omega_4)} = -i\omega_4 + \mu - \Sigma(\mu, \omega_4). \tag{47}$$

Since we are working at finite temperature, we have to do the Matsubara sum. However, notice that the $i\omega + \mu$ term has no poles, so when we evaluate the sum using the contour integration prescription, contribution of this part vanishes and we are left with,

$$\mathcal{F}_{\text{amp}}(\tau_1, \tau_2, \tau_3, \tau_4) = \int_0^\beta d\tau_0 \, \mathcal{F}(\tau_1, \tau_2, \tau_3, \tau_0) \Sigma(\mu, \tau_{40}),$$

$$\mathcal{F}_{\text{amp}}(\tau_1, \tau_2, \tau_3, \tau_4) = J^2 \int_0^\beta d\tau_0 \, \mathcal{F}(\tau_1, \tau_2, \tau_3, \tau_0) G(\tau_{40})^{q/2-1} G(-\tau_{40})^{q/2}. \tag{48}$$

Now we can convert the $\tau$ integrals to $\theta$ integrals via appropriate scaling and we get the contribution from the contact diagram to be,

$$S_c \sim \frac{q(\beta \mathcal{J}_{eff})^3}{(2\pi)^2}, \tag{49}$$

where,

$$\mathcal{F}_{\text{amp}}(\tau_1, ..., \tau_4) \sim \frac{\beta \mathcal{J}_{eff}}{2\pi q \beta}. \tag{50}$$

In terms of the $\theta$ variable we have,

$$\mathcal{F}(\theta_i, \theta_j, \theta_a, \theta_b) \sim \beta \mathcal{J}_{\text{eff}} G(\theta_{ij}) G(\theta_{ab}), \tag{51}$$

and in the large $q$ limit, for large but finite $\beta \mathcal{J}_{\text{eff}}$,

$$(G(\theta_{ab}))^{\frac{q}{2}} (G(-\theta_{ab}))^{\frac{q}{2}} \sim \frac{1}{(\beta \mathcal{J}_{\text{eff}})^2 \sin^2\left(\frac{\theta_{ab}}{2}\right)}. \tag{52}$$

Here we have put $v = 1$ inside the sine function which is consistent to the leading order with $v \to 1$ as $\beta \mathcal{J}_{eff} \to \infty$. As a consequence the $(\beta \mathcal{J}_{\text{eff}})^2$ coefficient of the contact diagram as well as the amputated four point function cancels out due to the $(\beta \mathcal{J}_{\text{eff}})^2$ appearing in the denominator of (52).

The planar six-point diagram, on the other hand, is given by the product of three amputated four point functions hence, when we take the product and convert the $\tau$ integrals to $\theta$ integrals in (46), we get,

$$\mathcal{S}_p \sim \frac{(\beta \mathcal{J}_{\text{eff}})^3}{q^3 (2\pi)^6}. \tag{53}$$

Taking the ratio of $S_c$ with $\mathcal{S}_p$ we see that,

$$\frac{S_c}{\mathcal{S}_p} \sim (2\pi)^4 q^4. \tag{54}$$

So in the large $q$ limit as one can easily see that the contact diagram is far more dominant compared to the planar ones and hence it is justified to consider the contribution of the contact diagrams only.

## 4.3 The six point function

Although one can get an analytic answer for the enhanced four point contribution slightly away from the conformal limit, calculation of the full six point function becomes somewhat messy to carry out analytically. We therefore compute the six point function using numerical methods.

Let us first summarize the results, we will then we state all the relevant values used in carrying out these computations.

- We first compute the six point contribution with three $h = 2$ modes, for a fixed $\mu\beta$, keeping all the time arguments separate and then we take the short time limit $\theta_2 \to \theta_1$, $\theta_4 \to \theta_3$, etc. We then change the value of $\mu\beta$ and compute the correlator again. We see that the six point function decreases in this limit as $\mu\beta$ is increased, while keeping $\beta\mathcal{J}$ fixed at some large value.

- We can instead first take the triple short time limit and then carry out the integrals numerically for all the possible nonconformal contributions, *i.e.*,

$$\mathcal{F}_{h=2}\mathcal{F}_c\mathcal{F}_c, \quad \mathcal{F}_{h=2}\mathcal{F}_{h=2}\mathcal{F}_c, \quad \mathcal{F}_{h=2}\mathcal{F}_{h=2}\mathcal{F}_{h=2}. \tag{55}$$

We find that among the three terms in (55), the first contribution seems to vanish to all orders in $\mu\beta$ expansion, on the other hand the remaining two terms, although are small, have contribution at the same order, namely $O\big((\mu\beta)^2\big)$. These results will get corrected as we go to higher orders.

- To benchmark the code we compute $\lambda_{11k}^{(1)}$ (as was done in [41] for all three conformal modes)[5] for the contact diagrams and plot it against $k$, where for large $k$, $h_k = 2k+1+2\Delta + O(1/k)$. We find the similar fall off behavior at large $k$.

Let us now look at some details of the analysis. One of the things that we have to keep in mind is that we are slightly away from the conformal limit because we have turned on a small $\mu\beta$. We need to be careful while working with the conformal modes. Due to explicit scale in the theory, the modes may not be conformal anymore. In other words, the normalized four point contributions of these modes may not be a function of only the cross ratio $\chi$ anymore. However, for small $\beta\mu$, the conformal perturbation theory makes sense and within this limit using the conformal basis is justified.

If we recall the eigenvectors of the Kernel then we see that,

$$\Psi(\theta_1, \theta_2) = e^{\frac{\mu\beta}{2\pi}(\theta_{12})} e^{in\frac{\theta_1+\theta_2}{2}} \psi_n(\theta_{12})$$

and to obtain the conformal modes one has to go to the IR, do the sum over $n$ in the four point function to obtain the sum over integer values of $h$ as well as the integral over the principle continuous series. One then deforms the contour to pick up the poles at $k_c = 1$, eigenvalue of the kernel in the conformal limit. In the IR limit the exponential $\mu\beta$ factor becomes equal to 1, but since it has no $n$ dependence it plays no role when we carry out the sum over $n$. Therefore,

---

[5]$\lambda_{nmk}^{(1)}$ defines the bulk cubic coefficient for Contact diagrams of operators with conformal dimensions $h_n = 2n + 1 + 2\epsilon_n$, $h_m = 2m + 1 + 2\epsilon_m$ and $h_k = 2k + 1 + 2\epsilon_k$. $\left(\epsilon_j = \frac{1}{q}\frac{j(2j+1)+1}{j(2j+1)-1}\right)$ with $j = n, m, k$

slightly away from the conformal limit we will have (small $\mu\beta$),

$$\frac{\mathcal{F}_{h\neq 2}(\theta_1,\theta_2,\theta_3,\theta_4)}{G(\theta_{12})G(\theta_{34})} = e^{\frac{\mu\beta}{2\pi}(\theta_{12}+\theta_{34})}\sum_{m=1}^{\infty}c_m^2\chi^{h_m}\,{}_2F_1(h_m,h_m,2h_m,\chi) \quad (56)$$

$$= \sum_{m=1}^{\infty}c_m^2\chi^{h_m}\,{}_2F_1(h_m,h_m,2h_m,\chi) \;+\; \frac{\mu\beta(\theta_{12}+\theta_{34})}{2\pi}\sum_{m=1}^{\infty}c_m^2\chi^{h_m}\,{}_2F_1(h_m,h_m,2h_m,\chi)$$

$$+\left(\frac{\mu\beta}{2\pi}\right)^2\frac{(\theta_{12}+\theta_{34})^2}{2!}\sum_{m=1}^{\infty}c_m^2\chi^{h_m}\,{}_2F_1(h_m,h_m,2h_m,\chi)+\cdots.$$

The above expression breaks conformal invariance and this is the four point function we will be working with away from the conformal limit.

Since $\beta\mathcal{J}_{\text{eff}}$ appears as an overall factor we strip off this factor and look at the integrals only. For small $\mu\beta$ this factor is large but finite. We will keep $\mu\beta$ fixed at $\mu\beta \sim 7.4\times 10^{-4}$.

## 5 The Short time and OTO behavior of the Six point function

In this section we will study short time limit of the six point function as well as its out-of-time ordered behaviour. In both cases the expressions are quite involved and we had to take recourse to numerical methods. We will first look at the short time limit.

### 5.1 Short time limit of the six-point function

Let us first look at the short time limit of the six point function. Since the terms we will be looking at are a part of the non-conformal piece, we will first compute it by keeping all six times different. We will then take the limit in which one of the temporal variable is approaching another, for example $\theta_2 \to \theta_1$, while holding rest of the time variables fixed. In this limit we will study the behaviour of the six point function. We do not have analytic control over this computation and hence we take a recourse to the numerical methods. Figure 2 contains the plot of numerical evaluation of the behavior of the six point, involving only the $h=2$ modes, as a function of $\theta_1$ in the short time limit, $\theta_2 \to \theta_1$ while keeping $\theta_3 = \theta_4$, and $\theta_5 = \theta_6$ fixed. In the triple short time limit, it is easy to see that the six point function vanishes.

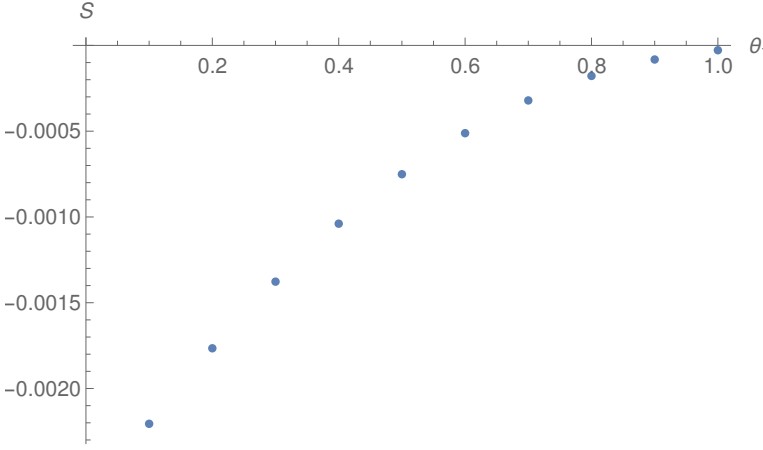

Figure 2: The plot of the six point function $S$ in the short time limit $\theta_2 \to \theta_1$. In this figure we have chosen arguments $\theta_3 = \theta_4 = \pi$, and $\theta_5 = \theta_6 = 5\pi/3$.

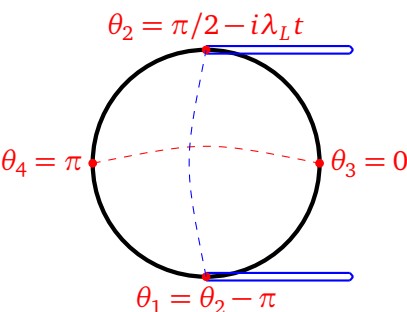

Figure 3: The 2-fold Schwinger-Keldysh contour characterising $\langle \psi_i(t)\psi_j(0)\psi_i(t)\psi_j(0)\rangle$

While the contribution to the non-conformal piece coming from the product of $h = 2$ modes is shown in Fig. 2, computing the contribution for one or two $h \neq 2$ modes requires some care because in this case the planar diagrams may have enhanced contribution but in the large q limit, it will still be subdominant. In a similar way it can be checked that the planar contribution is finite by analysing behavior of $\mathcal{F}_{\mathrm{amp}}(\theta_1, \cdots, \theta_4)$ as a function of its arguments. However, we do not explicitly compute this because we do not need it in our analysis.

## 5.2 Maximal Lyapunov from six-point OTOC

Before discussing the details, let us emphasize that even though we have analytic control over the ingredients, thanks to the large $q$-expansion, the calculation of the Euclidean correlator involves performing multi-variable integrations over the Euclidean time. The resulting integral, for the 6-point correlator, is a function of six Euclidean times. Ideally, one needs to first obtain this function of six variables, and subsequently carry out the analytic continuation corresponding to the Lorentzian correlator that one wants to compute. However, this is a challenging task in general, and it is even more so when one lacks an analytic expression for the Euclidean 6-point correlator. In view of this, we will discuss below a simple-minded numerical approach that appears to capture the correct Lyapunov behaviour.

Before considering the 6-point function, let us concentrate on the 4-point function evaluated in [4]. It is known to produce to well-established maximal value $\lambda_L = 1$, in units of $T = 1/(2\pi)$. The assignment of the time arguments in the alternating configuration, in [4], was set to be:

$$\theta_2 = \frac{\pi}{2} - it, \quad \theta_1 = \theta_2 - \pi, \quad \theta_3 = 0, \quad \theta_4 = \pi \,. \tag{57}$$

This implies that both $\theta_1$ and $\theta_2$ are analytically continued to real time and from their assignment on the thermal circle we find figure 3 as the corresponding Schwinger-Keldysh contours. Clearly, in this case, one computes a two Lorentzian time-folded correlator which corresponds to the true 4-point OTOC. The ordering of the imaginary time co-ordinates do not matter when analytically continuing to real time. There the ordering is fixed by epsilon prescription. The fact that the above chosen configuration yields the expected value of the maximal Lyapunov exponent, suggests that the information of the epsilon prescription can be translated to the configuration we choose. Furthermore, note that, even though the general 4-point OTOC is a function of four real-time variables, the choice in (57) simplifies this dependence to only one real-time variable, denoted by $t$. A similar statement holds for the Euclidean correlator.

Given the above choice of time-assignments, we perform the following tasks: (i) We numerically generate Euclidean data points, this yields a function $F_4(\theta)$, where $F_4$ is the Euclidean four-point correlator and $\theta$ is the only Euclidean time variable, with our proposed assignment in (57). (ii) We numerically fit the data with a guess-function. By trial and error, and motivated

by simplicity, we have used a cosine function as the fitting function, in which the frequency and the amplitude of the fitting function are found out as a result of the numerical fitting. (iii) We read off the frequency to be the corresponding Lyapunov exponent, since

$$\cos(\lambda_L \theta) \rightarrow ()_1 e^{\lambda_L t} + ()_2 e^{-\lambda_L t} \,, \tag{58}$$

where $()_{1,2}$ are overall constants. In the large time limit, such a term is dominated by the $e^{\lambda_L t}$, and therefore the Lyapunov is obtained from the frequency of the Euclidean periodic function. We have explicitly checked that, the above numerical implementation does reproduce $\lambda_L = 1$, within acceptable numerical accuracy, for the OTOC considered in [4]. This is not a water-tight argument that such an analysis will work for arbitrary cases, but, in absence of a better method, worth exploring. Furthermore, note that one does not need the complete information about the frequency spectrum of the Euclidean correlator, but only the dominant one, to extract the maximum Lyapunov exponent. This may provide a justification of why this method should work, however, we do not have any further reasons to offer at this point.

We will subsequently use a similar numerical method to compute the 6-point OTOCs, both with two time-folds and three time-folds. Finally, we will fit the data and extract the maximum Lyapunov exponent.

### 5.2.1  2-fold Schwinger-Keldysh contour

First we compute the quantity:

$$\langle \psi_i(\tau)\psi_j(0)\psi_k(0)\psi_i(\tau)\psi_j(0)\psi_k(0)\rangle \,,$$

which corresponds to the OTOC with two time-folds and $\tau$ denotes the Euclidean time. Thus, this is not a truly 6-point OTOC, since the out-of-time physics is captured by a 4-point OTOC that lies inside the 6-point correlator. Motivated by our discussion above, in this case, we assign:

$$\theta_1 = \frac{\pi}{6} + \tau, \ \theta_2 = \theta_1 + \frac{3\pi}{2}, \ \theta_3 = \frac{\pi}{3}, \ \theta_4 = \theta_3 + \frac{3\pi}{2}, \ \theta_5 = \frac{\pi}{2}, \ \theta_6 = \theta_5 + \frac{3\pi}{2} \,, \tag{59}$$

and we choose $\tau = -0.5, -0.4, ...., 0.5$. Thus we have,

$$\frac{\theta_1 + \theta_2}{2} \ = \ \pi + \tau \,.$$

Note that, within the regime of the $\tau$-variable: $\tau \in [-0.5, +0.5]$, we ensure the ordering: $\theta_1 < \theta_3 < \theta_5$ and therefore $\theta_2 < \theta_4 < \theta_6$, by choosing the configuration in (59). This ordering ensures that there are no operator crossing and the corresponding correlator remains a two-folded OTO correlator, for the regime of $\tau$ considered above. We note that this is only a representative configuration, however, the qualitative physics does not appear to significantly change.

We set the following values for different quantities:

$$q = 10, \ \Rightarrow \ a = \frac{(\mu\beta)^2}{2} = 0.001 \,.$$

The 2-fold contour, corresponding to the assignment in (59), is shown in figure 4. Now we compute the numerical data, as a function of $\tau$, the result of which is shown in figure 5. To this data, we fit a function of the form:

$$c_1 \ + \ c_2 \cos\left(\lambda_L \left(\frac{\theta_1 + \theta_2}{2} + \pi\right)\right) \ \equiv \ c_1 \ + \ c_2 \cos(2\pi\lambda_L + \lambda_L \tau). \tag{60}$$

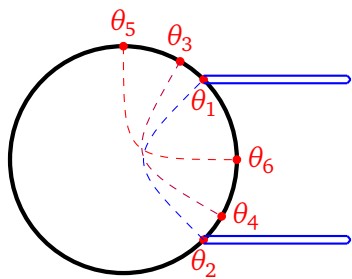

Figure 4: The 2-fold Schwinger-Keldysh contour for six-point OTOC

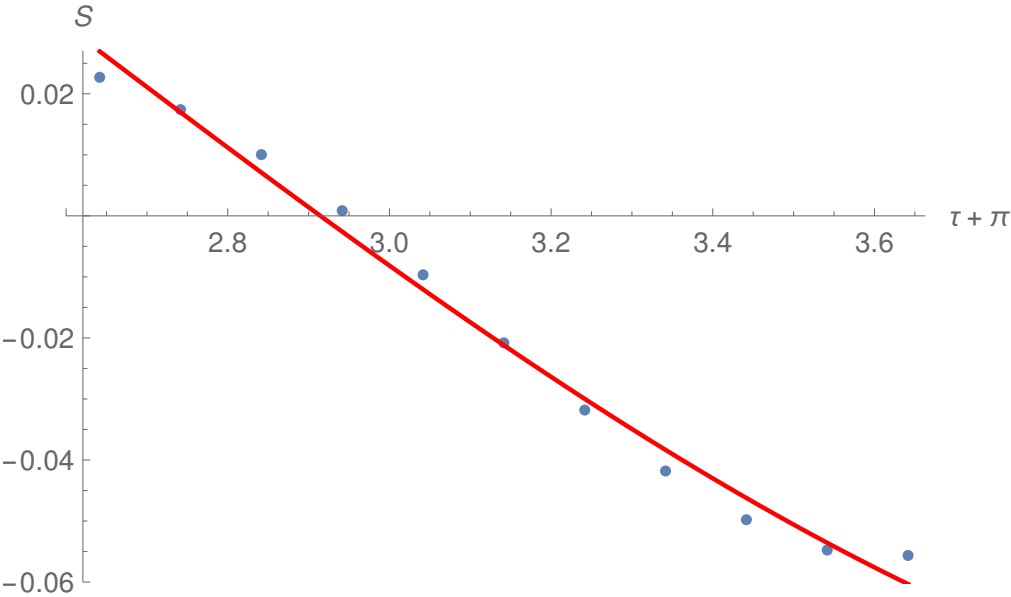

Figure 5: Cosine fit to the data with step-size=0.05. $\lambda_L = 0.823131$

Now, under analytic continuation $\tau \to it$, we end up with a real valued function. For the step size 0.05, the data can be fitted to the function in (60) with the following values of the coefficients and frequency:

$$c_1 = 0.0327781 \; ; \quad c_2 = -0.121711 \; ; \quad \lambda_L = 0.823131 \; . \tag{61}$$

Clearly, $\lambda_L$ is not far from the maximal value, but it is not sufficiently close.

### 5.2.2 3-fold Schwinger Keldysh contour

Now let us consider the quantity:

$$\langle \psi_i(\tau)\psi_j(0)\psi_k(\tau)\psi_i(0)\psi_j(\tau)\psi_k(0) \rangle \, ,$$

which is a true 6-point OTOC. The corresponding assignment of the Euclidean coordinates are given by

$$\theta_1 = \frac{\pi}{6} + \tau, \; \theta_2 = \frac{7\pi}{6}, \; \theta_3 = \frac{\pi}{3}, \; \theta_5 = \theta_1 + \frac{\pi}{3}, \; \theta_4 = \theta_5 + \frac{5\pi}{6}, \; \theta_6 = \frac{3\pi}{2} \; . \tag{62}$$

This implies:

$$\frac{\theta_1 + \theta_4 + \theta_5}{3} = \frac{2\pi}{3} + \tau \ .$$

This configuration corresponds to the following 3-fold contour: As before, it is straightforward to check that, for the range of $\tau \in [-0.5, +0.5]$, the configuration in (62) respects the ordering: $\theta_1 < \theta_3 < \theta_5$ and therefore $\theta_2 < \theta_4 < \theta_6$. Thus, the 3-folded OTO correlator remains a 3-folded one, for the entire variation of $\tau \in [-0.5, +0.5]$.

The numerical result is plotted in figure 7, along with the fitting curve, of the form:

$$c_1' + c_2' \cos\left(\lambda_L \left(\frac{\theta_1 + \theta_4 + \theta_5}{3} + \frac{4\pi}{3}\right)\right) \ \equiv \ c_1' + c_2' \cos\left(2\pi\lambda_L + \lambda_L \tau\right) \ .$$

From the fit we obtain the following values of the coefficients and frequency:

$$c_1' = -0.193813 \ , \quad c_2' = 0.191047 \ , \quad \lambda_L = 1.01853 \ . \tag{63}$$

Once again, the value of the corresponding Lyapunov is close to the maximal one.

Before concluding this section, let us note that our calculations are done near the conformal point and therefore it is not surprising that the corresponding Lyapunov exponent is numerically close to its maximal value. However, a few comments are in order.

Firstly, note that all our calculations are performed in the Euclidean description. By construction, the Euclidean time $\tau \leq \beta$, where $\beta$ is the inverse temperature. At best, we can ascertain $\tau \sim \mathcal{O}(\beta)$. Naively, within this regime, the analytic continuation to the Lorentzian time, *via* $\tau \to it$, should capture physics at time-scales $t \sim 1/T \sim t_d$, where $t_d$ is the dissipation time-scale. Therefore, our method is expected to capture the early-time physics that is still close to the physics at the dissipation time-scale.

It is therefore quite interesting that, at a pragmatic level, both the two-fold and the three-fold OTO correlators seem to capture an exponential growth even at this early time-scale. Moreover, the three-folded OTO seems to suggest a maximal Lyapunov exponent already at these time-scales, while the 2-fold OTOC (which is a four-point OTOC, fused with a two-point correlator; and not a truly 6-point OTOC) suggests an exponential growth, slightly below the maximal behaviour.

Naively, such a behaviour is not completely unexpected, since a higher point time-ordered correlator can decay faster than a lower point time-ordered correlator. Stated simple, slightly after the dissipation time-scale, a 4-point function $\langle \phi \phi \phi \phi \rangle \sim \langle \phi \phi \rangle \langle \phi \phi \rangle \sim e^{-2t/t_d}$, while $\langle \phi \phi \phi \phi \phi \phi \rangle \sim \langle \phi \phi \rangle \langle \phi \phi \rangle \langle \phi \phi \rangle \sim e^{-3t/t_d}$, assuming that the two-point function decays as $\langle \phi \phi \rangle \sim e^{-t/t_d}$. Therefore, as compared to the 4-point OTOC, the 6-point OTOC can already begin showing a chaotic growth that is closer to the large time behaviour of the corresponding correlator. In general, therefore, an $n$-point function, for sufficiently large $n$, can display the maximal Lyapunov growth at very early times. Of course, these statements are only plausible

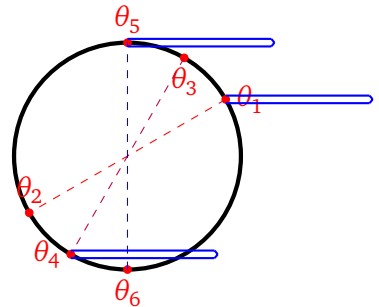

Figure 6: The 3-fold Schwinger-Keldysh contour characterising six-point OTOC.

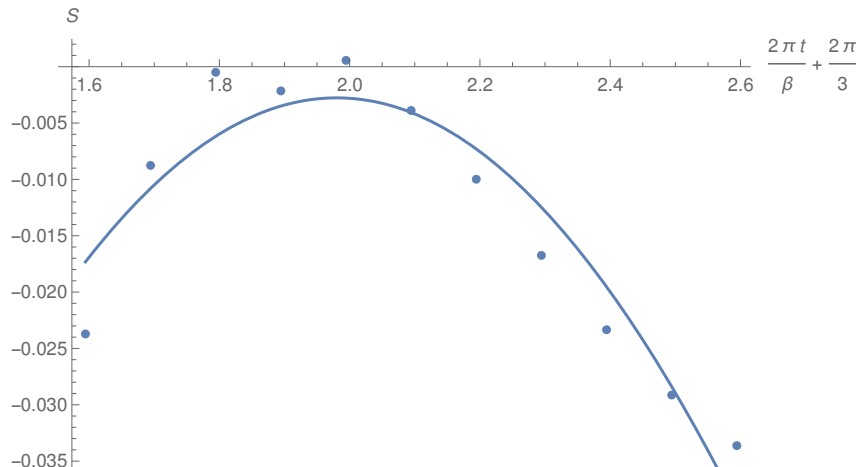

Figure 7: Six point OTOC with 3-fold Schwinger-Keldysh contour. Step-size= 0.05, $\lambda_L = 1.01853$

ones, based on the limited evidence that we have gathered for this particular example. To gain some confidence in this plausibility argument, we studied behaviour of the six point OTOC with two fold and three fold contour with same ordering but for a couple of slightly differing choices of the argument. We find that the Lyapunov index is reasonably robust under such variations which further justifies our argument. It will be very interesting to explore this in more detail by considering different arguments of the OTOC as well as considering higher point OTOC correlators.

# 6 Conclusion

We have computed the fermion six point function in the SYK model with complex fermions in the presence of a non-vanishing chemical potential. We then took triple short time limit of this correlation function so that it appears as a three point function of fermion bilinears. We show that the three point function of fermion bilinears, for $h \neq 2$ modes, have the scaling property of conformal field theory three point function, as is expected as a generalisation of the results of [41] to the complex fermion case. Like in [41], we find that the contribution of the contact three point graphs in the large $q$ limit is subleading compared to that of the planar graphs.

We also compute three point function of fermion bilinears for the $h = 2$ mode. This mode is known to break the conformal invariance of the SYK model, both spontaneously as well as explicitly. This mode is known to exhibit chaotic behaviour with the Lyapunov exponent $\lambda_{\rm L}$ that saturates the chaos bound. The three point function of bilinears in this case has a behaviour different from those of the conformal, *i.e., $h \neq 2$* modes. In this case we find that in the large $q$ limit, the contribution of the planar graphs is subleading compared to the contact graphs.

We have also explored the out-of-time-order physics of the associated six-point correlation function, upon taking the analytic continuation of the Euclidean result. As we have emphasized before, our approach is expected to extract an early-time Lyapunov growth in the corresponding OTOC. It is rather interesting that, for a higher point OTOC (*e.g.* the 6-point one), this early-time Lyapunov growth already features closely a maximal growth, while the 4-point OTOC already hints at the chaotic growth behaviour with a close to maximal Lyapunov behaviour. While there is no apparent inconsistency with the "maximally braided, $k$-OTO" correlation function studied in [42], it raises the possibility that the late-time Lyapunov growth

may already be pragmatically distillable at early-time physics, provided one explores an $n$-point OTOC, for sufficiently large $n$. Evidently, an explicit calculation of an $n$-point OTOC, for larger and larger values of $n$ becomes more and more difficult. So, one is not able to bypass the difficulty of understanding large-time physics, by simply computing a sufficiently high-point OTOC at early-times. These points deserve a better and more thorough scrutiny, and perhaps a natural point to explore is to consider the formal limit of $n \to \infty$. We hope to come back to this aspect in future.

As a further future direction to explore, coming back to the model at hand, since the couplings of the SYK model are chosen from random gaussian distributions, it is also tempting to ask if one can apply techniques of stochastic quantisation to reconstruct the bulk description. We hope to report on this soon.

**Acknowledgments:** We would like to thank Subhroneel Chakrabarti for participating in the initial stages of this work and for many discussions. RB would also like to thank Kasi Jaswin for important discussions related to the OTOC behavior. We would like to thank referees for their constructive remarks which helped improve the results and presentation considerably. This research was supported in part by the International Centre for Theoretical Sciences (ICTS) during a visit for participating in the program - AdS/CFT at 20 and Beyond (Code: ICTS/adscft20/05) during the course of this work.

# A  Parameters appearing in eq.(22)

In this appendix we collect the expressions of the constants that appear in six point amplitude.

$$c_n = \frac{2q}{(q-1)(q-2)\tan(\pi\Delta)} \frac{(h_n - \frac{1}{2})}{\tan\left(\frac{\pi h_n}{2}\right)} \frac{\Gamma^2(h_n)}{k'_A(h_n)\Gamma(2h_n)} \, , \tag{64}$$

$$\tilde{c}_n = \frac{2q}{(q-1)(q-2)\tan(\pi\Delta)} \frac{(h_n - \frac{1}{2})}{\cot\left(\frac{\pi h_n}{2}\right)} \frac{\Gamma^2(h_n)}{k'_S(h_n)\Gamma(2h_n)} \, , \tag{65}$$

$$\xi_n = b^q \pi^{1/2} \frac{\Gamma(1-\Delta+\frac{h_n}{2})\Gamma(\frac{1}{2}-\frac{h_n}{2})\Gamma(\Delta)}{\Gamma(\frac{1}{2}+\Delta-\frac{h_n}{2})\Gamma(\frac{h_n}{2})\Gamma(\frac{3}{2}-\Delta)} \, , \tag{66}$$

$$\tilde{\xi}_n = b^q \pi^{1/2} \frac{\Gamma(\frac{1}{2}-\Delta+\frac{h_n}{2})\Gamma(1-\frac{h_n}{2})\Gamma(\Delta)}{\Gamma(\Delta-\frac{h_n}{2})\Gamma(\frac{1}{2}+\frac{h_n}{2})\Gamma(\frac{3}{2}-\Delta)} \, . \tag{67}$$

# B  Four point function used in the computation of the six point function

In this appendix we will present the explicit form of various elements that go into the calculation of the 6-point correlator. The basic object of interest is the following function:

$$S(t_a, t_b; t_1, t_2, t_3, t_4, t_5, t_6) = \frac{1}{\sin^2\left(\frac{t_a-t_b}{2}\right)} \mathcal{F}(t_1, t_2, t_a, t_b; a) \times$$
$$\mathcal{F}(t_3, t_4, t_a, t_b; a)\mathcal{F}(t_5, t_6, t_a, t_b; a) \, , \tag{68}$$

where

$$
\begin{aligned}
\mathcal{F}(&t_1, t_2, t_a, t_b; a) = \\
&\cot\left(\frac{t_1 - t_2}{2}\right)\cot\left(\frac{t_a - t_b}{2}\right)\left(-p_1(t_1 - t_b, a) - p_1(t_2 - t_a, a) + p_1(t_2 - t_b, a) + p_1(t_1 - t_a, a)\right) \\
&+ \cot\left(\frac{t_a - t_b}{2}\right)\left(p_2(t_1 - t_a, a) - p_2(t_1 - t_b, a) + p_2(t_2 - t_a, a) - p_2(t_2 - t_b, a)\right) \\
&- \cot\left(\frac{t_1 - t_2}{2}\right)\left(p_2(t_1 - t_a, a) + p_2(t_1 - t_b, a) - p_2(t_2 - t_a, a) - p_2(t_2 - t_b, a)\right) \\
&+ p_3(t_1 - t_a, a) + p_3(t_1 - t_b, a) + p_3(t_2 - t_a, a) + p_3(t_2 - t_b, a)\,,
\end{aligned}
\tag{69}
$$

with the following definitions:

$$
p_1(\theta, a) = \sum_{n=2}^{\infty} \frac{\cos(n\theta)}{n^2(n^2 - 1)(1 + a)}\,,
\tag{70}
$$

$$
p_2(\theta, a) = \sum_{n=2}^{\infty} \frac{\cos(n\theta)}{(n^2 - 1)(1 + a)}\,,
\tag{71}
$$

$$
p_3(\theta, a) = \sum_{n=2}^{\infty} \frac{\sin(n\theta)}{n(n^2 - 1)(1 + a)}\,.
\tag{72}
$$

Now, the corresponding six-point correlator is computed as a function of $\{t_1, \ldots, t_6\}$, by integrating over $t_a$ and $t_b$.

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
