# Peer review of "Chaotic Correlation Functions with Complex Fermions"

_SciPost Physics, doi:SciPost Phys. Core 4, 018 (2021)_

## Round 3 · Referee Report · Anonymous (Referee 1) · 2021-1-18

Report

I thank the authors for their careful consideration of my comments and for their efforts towards improvement. Unfortunately, I think section 5 (which contains the main novelties of this work) is still misguided. Here are some of my reasons:

(1) The OTOC in section 5.2.2 is strange: fermions with identical indices are being inserted far away from each other in real time (e.g. $\psi_i(t)$ and $\psi_i(0)$). This does not lead to a nice OPE limit in which one would usually apply the fermion bilinear technique. It would be better to have identical fermions inserted at nearby real times. It seems that much of the previous sections relies on an OPE limit. Are the authors sure that the integrals they evaluate are valid away from the OPE limit?

(2) Ignoring issue (1), let me discuss the Euclidean configuration (5.6) that the authors study to extract the Lyapunov growth (similar comments apply to (5.3)). I am still very skeptical about the procedure by which the Lyapunov exponent is extracted. I can see now that the authors really compute a Euclidean six-point function for a fixed Euclidean ordering, and then they explore how this function changes upon varying 3 of the 6 Euclidean insertion points simultaneously by equal and small amounts. It is important that $t$ is small (smaller than $\pi/6$ in the configuration (5.6)), because otherwise the insertion points would cross and the ordering would change. The new content is very helpful in this regard, and I agree now that upon analytic continuation this procedure does give information about the real-time OTOC in the correct configuration. However, this information is unfortunately limited to small values of $t$. The Lyapunov growth, on the other hand, concerns large values of $it$! The calculation therefore does not necessarily say anything about the chaos regime.

To illustrate this point, consider the following function:
\begin{equation}
\begin{split}
f(t) &= \frac{1}{100} \left\{ c_1' + c_2' \cos\left( 2(2\pi + t - \frac{2\pi}{3})\right)\right\} \\
& + \frac{99}{100} \left\{ c_1' + c_2' \left[ \cos(2\pi \lambda_L) - \lambda_L\sin(2\pi \lambda_L) \left( t- \frac{2\pi}{3} \right) -\frac{\lambda_L^2 \cos (2\pi \lambda_L)}{2}\left( t- \frac{2\pi}{3} \right)^2\right]\right\}
\end{split}
\end{equation}
The first line is the authors' fit (5.7) with a different (arbitrary!) Lyapunov exponent $2$. The second line is the Taylor expansion of the authors' fit up to second order. For small $t$, this function is essentially indistinguishable from the fit in figure 7. But for large imaginary $t$ my function $f(t)$ has a completely different behavior: it grows with Lyapunov exponent $2$ instead of $\lambda_L$.

To summarize, it seems to me that the authors' procedure can probably extract the behavior of the OTOC for very early times, but not for the times relevant to Lyapunov growth. The procedure is therefore not suitable for extraction of the Lyapunov exponent (unless one knows much more about the expected functional form of the six-point function that one should try to fit numerically).

At this point, unfortunately I have to reject the paper for publication in SciPost. The main results seem physically sensible, but they are not derived using trustworthy methods. After putting the results on firmer grounds, or at least phrasing them more conjecturally, I would suggest submission to a different journal.

---

## Round 3 · Referee Report · Anonymous (Referee 2) · 2021-1-19

Report

I would thank the authors for incorporating the changes suggested during previous review. However, there are still some revisions that need to be made by the authors. The paper can be considered for submission after incorporating them.

Requested changes

1- Throughout the section 5 the authors have used the variable $t$ to denote multiple quantities: Lorentzian time, Euclidean time and scaled Euclidean time (in the firgures 5 and 6). This reduces the clarity of the manuscript and I would suggest that the authors choose clearer notation.

2- Along the lines of questions raised by the other referee, I would like to point out that it will be useful to discuss and emphasize particular choices of signs that authors have made to ensure the particular ordering of operators. For example, in (5.3), despite $\theta_1+0.3>\theta_3$ (and also for $t=0.4,0.5$) the authors have committed to the chosen ordering in their numerics. I interpret the discussion following (5.1) about the ordering of operators as an attempt at explaining but I find that discussion even more confusing. Making it unambiguously clear as to why this ordering remains is important, especially for the case of 3-fold contour because otherwise a reader can easily get confused and deduce that for some values of $t$ the contour becomes 2-fold.

3- Going back to my previous comment on the symmetry factor. I still don't agree with the comments that the authors have made in response. I wasn't concern by the fully connected diagrams being 1/N suppressed with respect to the disconnected diagrams. I wish to once again point out that a diagram in which $\tau_1,\tau_3$ would be the arguments of the first four point function, the color contractions are such that in the 'planar limit' the diagram vanishes. (see the attached figure)

Attachment

---

## Round 3 · List of Changes

Ref 1 Comments:

  1. Following the referees comments number 1,2,3 and 4 we have rectifed the typos. Also changed the last paragraph of our introduction to include the contents of Section 5.

  2. Regarding the 5th comment we corrected the symmetry factor as was pointed out. Although the query regarding the suppression of such diagrams in the N- counting we think is already addressed in the beginning of page 8 para 2. The fully connected component of the six point function is indeed suppressed in N- counting w.r.t the diconnected diagrams. Both Contact and Planar diagrams are at the same order in 1/N.

Ref 2 Comments:

  1. Subsection 5.2 has been modified significantly. We have added arguments regarding the fact that what we compute is indeed OTO correlators. We draw comparison with the work of reference [4] to this end.

  2. The computation for both two fold and three fold OTOC have been included. We find that in both cases we get the same Lyapunov exponent within numerical accuracy.

  3. Two new diagrams and the plots corresponding to both two and three fold computations have been added.

---

## Round 4 · Referee Report · Anonymous (Referee 1) · 2021-4-16

Report

My previous points have been addressed. I am still somewhat skeptical about how much OTOC physics the numerical method captures (even at early times), but the phrasing of the manuscript is now sufficiently conjectural to reflect this. I would recommend publication of the article in its current form in the SciPost Physics Core journal.

---

## Round 4 · Referee Report · Anonymous (Referee 2) · 2021-4-29

Weaknesses

1- Poor presentation

Report

Based on the previous and current version of this manuscript and the presentation and the results discussed in this paper I would recommend it for publication in SciPost Core journal. I believe that some of the results in this paper might be useful for the community working on related topics.

---

## Round 4 · List of Changes

1. The notation for Euclidean time in Section 5 is now changed to \tau so that there is no confusion with the Lorentzian time ` t '.

  2. The arguments \theta_1,...,\theta_6 for the 2-fold case are now chosen more carefully so as to ensure that the operator ordering does not change when \theta_1 is changed.

  3. A comment in the Conclusion section is added to remark that the result we obtain is the Lyapunov growth at ``early time".

---

## Editorial Decision

published